# Exploring the Properties and Structure of Real Knowledge Graphs across Scientific Disciplines

## Abstract

1 Despite the recent popularity of knowledge graph (KG) related tasks and bench-
2 marks such as KG embeddings, link prediction, entity alignment and their use in
3 many domains, the structure and properties of real KGs are not well studied. In
4 this paper, we perform a large scale comparative study of 29 real KG datasets from
5 diverse domains such as the natural sciences, medicine, and NLP to analyze their
6 properties and structural patterns. Based on our findings we make recommenda-
7 tions regarding KG-based model development and evaluation. We believe that the
8 rich structural information contained in KGs can benefit the development of better
9 KG models across fields and we hope this study will contribute to breaking down
10 the existing data silos between different scientific disciplines (e.g., biomedicine,
11 ML/NLP, 'AI for Sciences').

## 1   Introduction

13 Recent years have been marked by an increased use of multimodal and structured datasets in the form
14 of knowledge graphs (KGs) to enhance applications in diverse scientific and technical disciplines
15 such as natural language processing (NLP), natural sciences and medicine, manufacturing and process
16 automation to name a few (Zou, 2020; Peng et al., 2023). The wide applicability of KGs is not
17 surprising: they are scalable data objects that store factual (i.e., with high degree of certainty)
18 information in the form of triples and allow for encoding both topological and semantic information.

19 The growing interest in using KG across various domains has led to a surge in the release of new KG
20 datasets: for example, 43% the 37 datasets available in the PyKEEN v1.10 [1] KG embedding library
21 have been published since 2020 (Ali et al., 2021b). Many other libraries – e.g., OGB (Hu et al., 2020,
22 2021), LibKGE (Ruffinelli et al., 2020; Broscheit et al., 2020) and PyTorch Geometric (Fey and
23 Lenssen, 2019) – consolidate multiple KG models in a central repository or provide tools for task-
24 specific benchmarking. Lastly, several recent studies focus on scalable benchmarking of KG related
25 tasks: for example Ali et al. (2021a) compare the performance of 21 KG link prediction models, Sun
26 et al. (2020) evaluate 12 embedding-based EA approaches on dedicated benchmark datasets, while
27 AlKhamissi et al. (2022) propose a KG-based framework for assessing the performance of pretrained
28 language models (PLMs) with the goal of achieving parity between PLMs and KGs.

29 Even with the abundance of KG datasets, benchmarking tools, and extensive large-scale model
30 comparisons across various KG tasks, to the best of our knowledge, there are no studies that address a
31 much more fundamental question, namely: *what properties and structure do real KGs have and how*
32 *do they compare to each other in terms of these properties?* We argue that a systematic approach of
33 analyzing KG properties (the goal of this paper) has the potential to inform algorithmic development

---

[1]Permalink: https://ezproxy.library.und.edu/login?url=https://github.com/pykeen/pykeen

Submitted to NeurIPS 2021 AI for Science Workshop.

Table 1: Datasets that were analyzed in this study. #E, #R, and #T denote the number of entities, relations and triples, respectively. $deg$ denotes the average degree of all the KG entities. $d$ denotes the KG density, shown in $\log$ scale (a lower column value implies a denser KG).

| # | dataset | # E | # R | # T | category | $deg$ | $-\log(d)$ |
|---|---------|-----|-----|-----|----------|-------|-----------|
| 1 | AristoV4 | 42,016 | 1,593 | 279,425 | biomed | 7 | 3.80 |
| 2 | BioKG | 105,524 | 17 | 2,067,997 | biomed | 20 | 3.73 |
| 3 | CoDExLarge | 77,951 | 69 | 612,437 | semantic | 8 | 4.00 |
| 4 | CoDExMedium | 17,050 | 51 | 206,205 | semantic | 12 | 3.15 |
| 5 | CoDExSmall | 2,034 | 42 | 36,543 | semantic | 18 | 2.05 |
| 6 | ConceptNet | 28,370,083 | 50 | 34,074,917 | semantic | 1 | 7.37 |
| 7 | Countries | 271 | 2 | 1,158 | society | 4 | 1.80 |
| 8 | CSKG | 2,087,833 | 58 | 4,598,728 | semantic | 3 | 5.98 |
| 9 | DB100K | 99,604 | 470 | 697,479 | semantic | 7 | 4.15 |
| 10 | DBpedia50 | 24,624 | 351 | 34,421 | semantic | 1 | 4.25 |
| 11 | DRKG | 97,238 | 107 | 5,874,257 | biomed | 60 | 3.21 |
| 12 | FB15k | 14,951 | 1,345 | 592,213 | semantic | 40 | 2.58 |
| 13 | FB15k-237 | 14,505 | 237 | 310,079 | semantic | 21 | 2.83 |
| 14 | Globi | 404,207 | 39 | 1,966,385 | biomed | 5 | 4.92 |
| 15 | Hetionet | 45,158 | 24 | 2,250,197 | biomed | 50 | 2.96 |
| 16 | Kinships | 104 | 25 | 10,686 | society | 103 | 0.01 |
| 17 | Nations | 14 | 55 | 1,992 | society | 143 | −1.01 |
| 18 | OGBWikiKG2 | 2,500,604 | 535 | 17,137,181 | semantic | 7 | 5.56 |
| 19 | OpenBioLink | 180,992 | 28 | 4,563,407 | biomed | 25 | 3.86 |
| 20 | OpenEA | 15,000 | 248 | 38,265 | semantic | 3 | 3.77 |
| 21 | PharmKG | 188,296 | 39 | 1,093,236 | biomed | 6 | 4.51 |
| 22 | PharmKG8k | 7,247 | 28 | 485,787 | biomed | 67 | 2.03 |
| 23 | PrimeKG | 129,375 | 30 | 8,100,498 | biomed | 63 | 3.32 |
| 24 | UMLS | 135 | 46 | 6,529 | biomed | 48 | 0.45 |
| 25 | WD50K | 40,107 | 473 | 232,344 | semantic | 6 | 3.84 |
| 26 | Wikidata5M | 4,594,149 | 822 | 20,624,239 | semantic | 4 | 6.01 |
| 27 | WN18 | 40,943 | 18 | 151,442 | semantic | 4 | 4.04 |
| 28 | WN18RR | 40,559 | 11 | 92,583 | semantic | 2 | 4.25 |
| 29 | YAGO3-10 | 123,143 | 37 | 1,089,000 | semantic | 9 | 4.14 |

across disciplines and empower the next generation of KG-based applications in NLP, biomedicine and other areas where KGs are used.

**Contribution and Scope**. In order to begin addressing the above question, we analyze the structure of KGs in terms of their network statistics, topology and relation types. Our large scale comparative study is based on 29 real KG datasets from diverse domains such as biology, medicine, and NLP. Towards our goal, we: (1) measure various KG properties (e.g., KG density, degree distribution); (2) analyze the KG structure in terms of the relational types and the KG topology; and (3) describe common/distinct structural patterns we observe KG datasets derived from fundamentally different underlying domains. Based on our findings, we make several recommendations for future model development and evaluation. Lastly, our primary goal is to analyze KG datasets and their properties along different dimensions, rather than benchmark downstream task-specific models on said datasets.

## 2 Application of Knowledge Graphs in Different Disciplines

**Notation**. For a given set of entities $E$ and a set of relations $R$, a knowledge graph $\mathcal{K} \subseteq K = E \times R \times E$ is a directed multi-relational graph that contains triples of the form $(h, r, t) \in \mathcal{K}$ in which $h, t \in E$ represent the head and tail entities and $r \in R$ is the relation between them. KG embedding models (e.g. TransE (Wang et al., 2014), DistMult (Yang et al., 2014)) learn latent vector representations of the entities $e \in E$ and relations $r \in R$ that best preserve the KG's structural properties.

**Natural Sciences and Medicine**. KGs are used in various biomedical applications (Nicholson and Greene, 2020; Ektefaie et al., 2023) and have recently found use in precision medicine (Chandak et al., 2023). In biology and medicine KGs typically describe the relationships between biomedical entities such as diseases, drugs, phenotypes, and regulatory pathways. They are a convenient tool for aggregating knowledge fragmented across publications, repositories, ontologies and databses (Chandak et al., 2023). KGs embeddings and link prediction find application in pharmaceutical applications (e.g. discovery and drug repurposing), clinical applications (e.g., disease diagnosis and treatment) and genomics (e.g., the study of phenotyping) (Morselli Gysi et al., 2021; Chandak et al., 2023; Wang et al., 2023). Natural science disciplines such as physics (Zou, 2020) and geology (Zhu et al., 2017) make use of multimodal data such as scientific literature and other natural language datasets to construct domain-specific KGs.

**ML and NLP**. In ML and NLP PLMs have gained immense popularity in recent years due to their impressive ability to process and generate human-like text. PLMs, such as GPT-4 (OpenAI, 2023) or Llama (Touvron et al., 2023) are able to generate answers to complex user queries on a variety of technical topics. However, these models are known to suffer from a lack of grounding of their outputs (in factual, common sense and domain specific knowledge) and from having difficulties in properly dealing with the meaning of inter-related concepts (Carta et al., 2023). Some approaches use factuality, common sense, physical and domain specific knowledge to mitigate the weaknesses of PLMs (Wei et al., 2021; Zamini et al., 2022; Hu et al., 2023); see (Yang et al., 2023) for a review. Many 'LMs as KBs' works, reviewed in (AlKhamissi et al., 2022), also use KGs for fine-grained evaluation of different aspects of PLMs such as their ability to recover factual knowledge or their consistency (Heinzerling and Inui, 2021). However, in none of those approaches the properties of the KGs at hand are considered.

**Other Areas of Science**. Many other domains such as cybersecuirty, finance, education, factory monitoring and process automation, geopolitics and combating human trafficking benefit from KGs and apply KG tasks such as EA and link prediction – Li et al. (2020); Zou (2020) provide a review of domain-specific KGs and their downstream use in these areas.

## 2.1 Knowledge Graph Datasets Analyzed

Among all the datasets in the various domains described above, we used a set of 29 KGs, which we list in Tab. 1 together with their summary statistics. We categorize them into three distinct groups:

1. **biomedical KGs** (count $=9$) which store facts related to biology and medicine such as relationships between genes, proteins or cellular pathways. Datasets in this group are typically derived from high quality public databases such as DrugBank (Wishart et al., 2018) and PubChem (Kim et al., 2016) – for construction details see (Zheng et al., 2021).

2. **semantic web KGs** (count $=17$) which incorporate knowledge extracted using the tools of the semantic web or analogous mechanism such RDF (Fensel, 2005). [2]. Many datasets in this category are derived from each other or share common sources – for example ConceptNet (Speer et al., 2017) is based, in part, on DBpedia (Auer et al., 2007), while CSKG (Ilievski et al., 2021) makes use of ConceptNet and Wikidata.

3. **societal KGs** (count $=3$) are a set of manually curated datasets that contain factual information about different domains such as geography and international relations We note that these KGs are conceptually similar to the ones in the biomedical domain in the sense that they are based on relationships between physical objects.

We use the datasets through the PyKEEN v1.10 software package (see details in the Appx.)

## 3 The Properties and Structure of Knowledge Graphs

The structural characteristics of KGs play an important role in the applicability and the performance of various tasks such as KG embeddings, link prediction and reasoning. For example, KG properties such as relation type (e.g., inverse, symmetric), cardinality and statistics affect the KG connectivity

---

[2]https://www.wikidata.org/wiki/Wikidata:Database_download

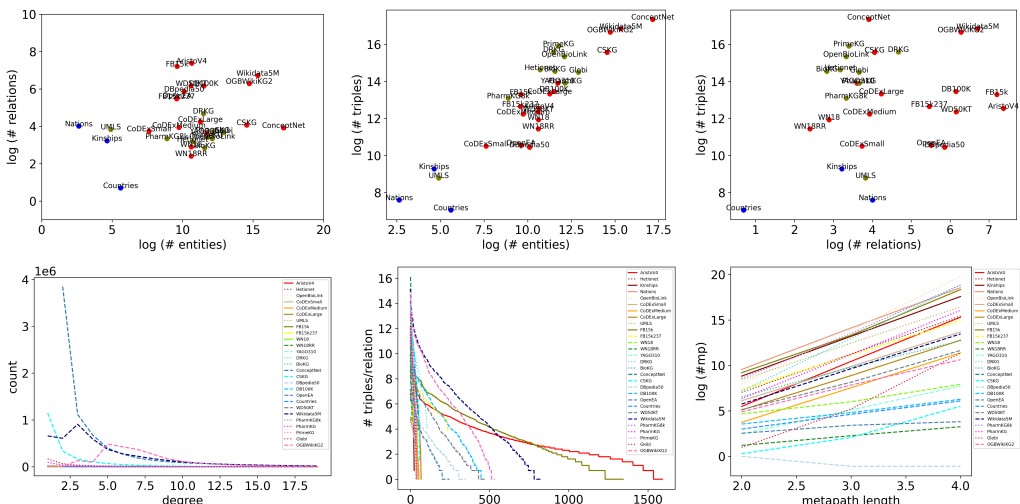

Figure 1: (**Top**) Relations between (left) number of entities vs. number of relations; (center) number of entities vs. number of triples; (right) number of relations vs. number of triples in different KGs. Biomedical, semantic web and societal datasets are colored in resp. olive, red, and blue. All axes are $log$-scale. (**Bottom**) (left) Degree distribution; (center) number of triples per KG relation, $log$ scale; (right) metapath (mp) length distribution on the $y$-axis, $log$ scale.

patterns which get encoded in the KG node and relation embeddings and used in downstream models. The effect releation types have was demonstrated by Toutanova and Chen (2015) who first described the 'inverse relation problem' in KG link prediction: essentially, an information leakage between the train and test splits due to the presence of *inverse* relations in the training dataset splits. They identified the issue in the FB15k dataset and released its updated version FB15k-237, both commonly used in NLP benchmarks. Moreover, the inference abilities of KG embedding algorithms vary by relation type and carinality. For example, TransE cannot model symmetric and one-to-many relations well due to its scoring function $f(r, h, t) = -||h+r-t||$. Similarly, the distance functions for DistMult and ComplEx cannot model composite relations. Cao et al. (2022) provide a detailed review.

In the ML and NLP literature semantic web datasets are prevalent and the performance of KG embeddings, EA models and link prediction (both embedding-based and Graph Neural Network (GNN)-based (Cucala et al., 2021) approaches) is often demonstrated only on a small set of semantic web datasets such as FB15k, WN18 and Wikidata5M. On the other hand, in the biomedical domain it has been widely accepted that semantic web datasets do not reflect the domain specific properties of biomedical KGs due to a variety of factors (Zheng et al., 2021; Breit et al., 2020). One such factor are the interaction effects in KGs: biomedical KGs have been found to be sparse, incomplete and containing richly structured ontological hierarchies with large interaction networks instead of capturing knowledge networks (e.g., FB15k) or hierarchical taxonomies (e.g., WN18)(Zheng et al., 2021; Breit et al., 2020). Another distinguishing characteristic of biomedical KGs is the nature of the entities stored in them – for example automatically created datasets such as OpenBiolink (Breit et al., 2020) contain large number of meta data entities and trivial biomedical entities which can interfere with KG embeddings and link prediction models as reported by Zheng et al. (2021).

**KG vs. Graph Structure**. KGs remain relatively unexplored as topological objects, partially because KG datasets were not readily available in libraries or benchmarks, such as PyKEEN or PyTorch Geometric until recent years. In contrast, as network analysis gained momentum in the 1990s, thanks to the growing availability of web, social and other real networks, the topology and numerical characteristics of directed and undirected graphs (the homogeneous counterparts of KGs) have been extensively studied. Multiple libraries and benchmarks have been established for the analysis of graph/network properties – examples include the general purpose network and graph mining library SNAP (Leskovec and Sosič, 2016) and the SuiteSparse matrix collection (Davis and Hu, 2011;

Kolodziej et al., 2019) which systematizes graphs together with their numerical characteristics and finds application in scientific computing research.

**Graph Structure in Existing Benchmarks**. Many KG benchmarks (Ali et al., 2021a; Widjaja et al., 2022) focus on standardizing model training, hyper parameter tuning or task-specific model evaluation. However, only a few benchmarks provide support for evaluation of KG tasks with respect to the underlying KG structural properties. One example is the KGxBoard framework for KG link prediction evaluation (Widjaja et al., 2022). KG link prediction performance is often measured using metrics (e.g., precision) averaged over a held-out set, however, as noted by Widjaja et al. (2022), single-score summary metrics cannot reveal exactly what the model has learned or failed to learn. To remedy this, the KGxBoard framework implements a fine-grained performance reporting per relation type. While in principle the KGxBoard framework offers support for multiple relation types, the authors did not perform a systematic evaluation of link prediction performance for multiple relation types or KG properties. In another benchmark, Ali et al. (2021a) evaluate 21 KG link prediction models with respect to four relational patters on 4 datasets (mix of semantic web and societal), however, they do not analyze the link prediction performance in the context of the KG relational distributions and properties. Lastly, Sun et al. (2020); Leone et al. (2022) benchmark KG heterogenity and its effect on EA performance.

# 4 Methodology and Results

**Methodology**. We perform a series of data analysis steps to measure various KG properties and structural dimensions across all datasets. Below we provide details on each KG dimension we considered, describe the experiments we conducted along each dimension and summarize empirical observations and findings. Unless otherwise noted, we used the datasets in their entirety (incl. train, test, and validation splits). The code from our experiments will be made available in the final version.

**Entities, relations and triples**. In Fig. 1 we show the relations between the total number of entities, relations and triples in each dataset. When plotting the number of entities vs. number of relations in Fig. 1 (top, left), we notice that on average semantic web KGs have more diversity in terms of the entity/relation count, while most biomedical KGs cluster at similar entity/relation count. This implies that regardless of the fact that many of the semantic web datasets are derivatives of each other, as mentioned in Sec. 2.1, they exhibit some diversity along these two dimensions. Notably, Fig. 1 (top, center) – plotting the number of entities vs. number of triples – shows the biomedical KGs cluster together with the exception of the PharmKG8K and UMLS datasetas. In the same panel (top right corner) we also observe that the largest KG datasets in our study are predominantly semantic web ones. Finally, Fig. 1 (top, right) shows that biomedical datasets exhibit less diversity in terms of the relation vs. triples count, in comparison the the semantic web datasets.

**Average degree and degree distribution** KG entities are connected to each other by directed edges (corresponding to the relations), hence an entity can have outgoing edges (or an out-degree) when it is the head entity in a triple and incoming edges (or an in-degree) when it is the tail entity in a triple. We sum the in-degrees and out-degrees to obtain the entity degrees analogously to the the way node degrees are computed in undirected graphs. A smaller average degree indicates that the KG is sparser. Fig. 1 (btm, left) shows the degree distribution (over all entitie). The average degree for each KG is also shown in Tab. 1. From this table we notice that the semantic datasets have some of the lowest average degrees across all datasets (e.g., ConceptNet with an average degree of 2). Among the semantic datasets FB15k has the singularly highest average degree, while its derivative FB15k-237 has a value that aligns with the rest of the semantic datasets. On the other hand, the average degrees of biomedical datasets are split into two groups: 5 of the biomedical datasets have average degrees in the low100's while PharKG, Globi, OpenBioLink, and BioKG have average degrees that are closer to the values of the semantic datasets. The two societal datasets Kinship and Nations demonstrate a significantly higher average degree than the rest of the KGs (see the Appx.) In Fig. 1 (btm., left) several datasets show a distinct degree distribution in the lower degrees. ConceptNet, CSKG have the highest number of low-degree nodes, while the degree distributions of Wikidata5M and OGBWikiKG2 are not as smooth, with oscillations at degrees 3 and 5, respectively.

**KG Density**. The KG density is computed as the ratio $|E|/|R|^2$ and like in homegeneous graphs higher density implies more sparsity (see Tab. 1, last column). The degree trends we described in the previous paragraph can also be traced along this dimension. Related to the KG density, we also quantify the KG connectivity by plotting the triples per KG relation in Fig. 1 (btm., center). The thick tails in the plot show that AristoV4, followed by FB15k, have a high number of relations with a low number triples per relation. Several other semantic datasets follow the same pattern, while none of the biomedical datasets do, with the exception of DRKG.

**Relation cardinality**: Relation cardinality describes the numerical relationship between the possible head and tail entities of the relation (i.e., how many entities a relation can have as a tail or head). The possible types are: (i) one-to-one (1-1), (ii) one-to-many (1-M), (iii) many-to-many (M-M), (iv) many-to-one (M-1) (Widjaja et al., 2022). For illustration, the relation `GeneActivationGene` (from the OpenBioLink dataset) is M-M because various genes can activate multiple other genes. Fig. 2 plots the relation cardinality distribution for each dataset considered. We observe several different dataset profiles: **(i) 1-1 dominance**: In 15 of the 28 datasets in Fig. 2 the leading relation type is 1-1 including many semantic datasets. Overall, the number of 1-1 relations is more pronounced in the semantic datasets than in the biomedical datasets. **(ii) M-M dominance**: In 11 of the 28 datasets, the leading relation type is M-M. Biological datasets (e.g. BioKG, Hetionet, OpenBioLink, PharmKG8k) are dominated by M-M relations with the exception of PharmKG and Globi which exhibit a distinct profile. All societal datasets also fall in this category. **(iii) mixed cardinalities**: We observe that some of the most frequently used semantic datasets (FB15k, FB15k-237, Yago310) have a significant number of all 4 cardinalities unlike the rest of the analyzed datasets. **(iv) mixed profile with a 1-1 or M-M skew**: All biomedical datasets tend to be skewed towards M-M relations, except PharmKG and Globi. All semantic datasets tent to be skewed towards 1-1 relations, with the notable exception of Yago310.

**Relational patterns**. We considered four relation types described in the literature (Ali et al., 2021a; Toutanova and Chen, 2015). A relation $r \in R$ is: (i) **asymetric** if $(h, r, t) \in T \implies (t, r, h) \notin T$. (ii) **symmetric** if $(h, r, t) \in T \implies (t, r, h) \in T$. (iii) **inverse** to $r_{inv} \in R$ if $(h, r, t) \in T \implies (t, r_{inv}, h) \in T$ If there exists a $r' \in R$, s.t. $r' \neq r$ and $r'$ is inverse of $r$ then $r$ is an inverse relation. (iv) **composite** of two relations $r_1, r_2 \in R$ if $(x, r_1, y) \in T \land (y, r_2, z) \in T \implies (z, r, z) \in T$. Fig. 3 plots the relation pattern distribution for each dataset considered. Notably, some semantic datasets such as DB100K, OpenEA and DBpedia50 have a small amount of inverse relations which may affect benchmarking on these datasets in light of the 'inverse relation problem' discussed above. Across all datasets we observe dominance of anti-symmetric relations, with the exception of the societal datasets and some of the most frequently used benchmarking datasets in NLP, such as FB15 and WN18RR, which show presence of all 4 relation types. Some semantic and societal datasets have composite relations, while none of the biomedical do.

**Metapaths**: KG metapaths are widely used in the biomedical literature for assessing the connectivity of KGs and deriving insights about the clinical or biological relevance of interactions such as gene-gene or drug effects (Su et al., 2020; Fu et al., 2016; Himmelstein et al., 2017; Zhang et al., 2020). A metapath is defined as a sequence of relations separated by edge types (metanodes). For example, a metapath of length $\ell$ is of the form $e_1 \xrightarrow{r_1} e_2 \xrightarrow{r_2} \ldots \xrightarrow{r_{\ell-1}} e_\ell$ where each of $e_1, e_2,$ and $e_\ell$ belongs to a specific metanode. For example, in the Hetionet dataset(Himmelstein et al., 2017) the metanodes `Compound`, `Gene` and `Disease` form the metapath `Compound` $\xrightarrow{\text{binds}}$ `Gene` $\xrightarrow{\text{associates}}$ `Disease` of length 2. The number of metapaths of a given length provides a way for quantifying the level of relational composition without having explicit composite relations encoded in the KG. Interestingly, Cohen et al. (2023) test the reasoning abilities of PLMs by a prompting strategy that forces them to survey entity neighborhoods; although the authors do not put their work in the context of KG metapaths.

In practice, metapaths of length of greater than 4 are considered too long to make a significant contribution in link prediction task (Himmelstein et al., 2017; Fu et al., 2016). Fig. 1 (btm., right) compares the metapath length distribution over paths of length $2, 3$ and $4$ for all KGs (see Appx for additional details). From the figure we see that the biomedical datasets contain a significantly higher

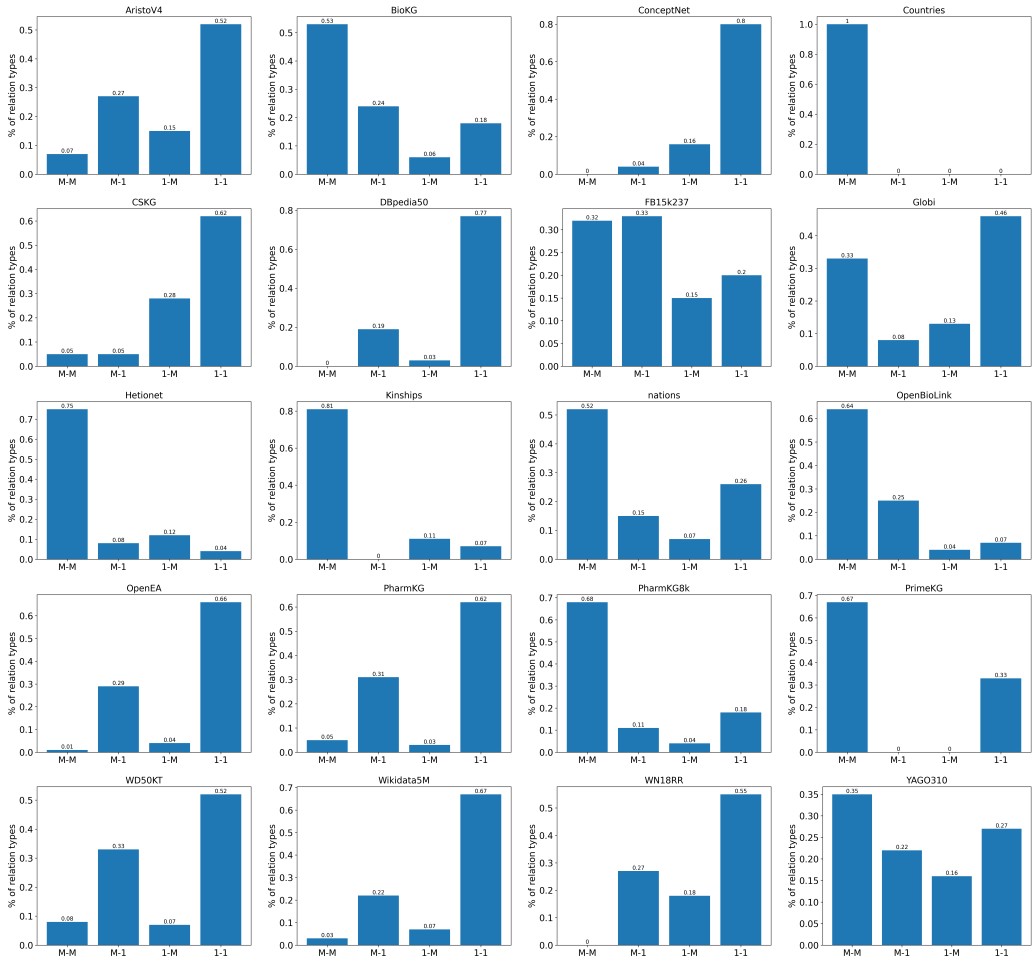

Figure 2: Distribution of relation cardinalities in different KGs. Each bar is marked with the # number of relations of the specified type. Due to space limits, the remaining datasets are shown in the Appx.

number of metapaths than the other types of datasets with the exception of Globi. However, among the semantic dataset FB15k and the societal datasets Kinship and Nations datasets exhibit a profile which is similar to the biomedical datasets.

## 5 Findings and Future Work

Below we highlight the key takeaways from our analysis and make several recommendations:

- Given the KG diversity observed in this study, we conclude that not all KGs are created equal. This has implications for model evaluation in all scientific domains KGs are used. Thus, we recommend that researchers consider a broader set of datasets beyond their target domain (biomedical, NLP, etc.) for their KG model development and evaluation. Fine-grained model evaluation – for example, as function of relation type, cardinality, KG density or degree distribution – has the potential to further drive the development of new KG-based models or inform model selection given the specific KG properties.
- Inverse relations are present in some datasets, including some released after the 'inverse relation' leakage problem was reported by Toutanova and Chen (2015). Given the implications of this problem in downstream applications, we recommend KG libraries and benchmarks consider adding tools for handling/removing inverse relations in order to bring visibility to the leakage problems.
- The overall negligible amount of composite relations in many datasets (including biological, semantic, and societal) is one interesting observation that merits further analysis. Composite

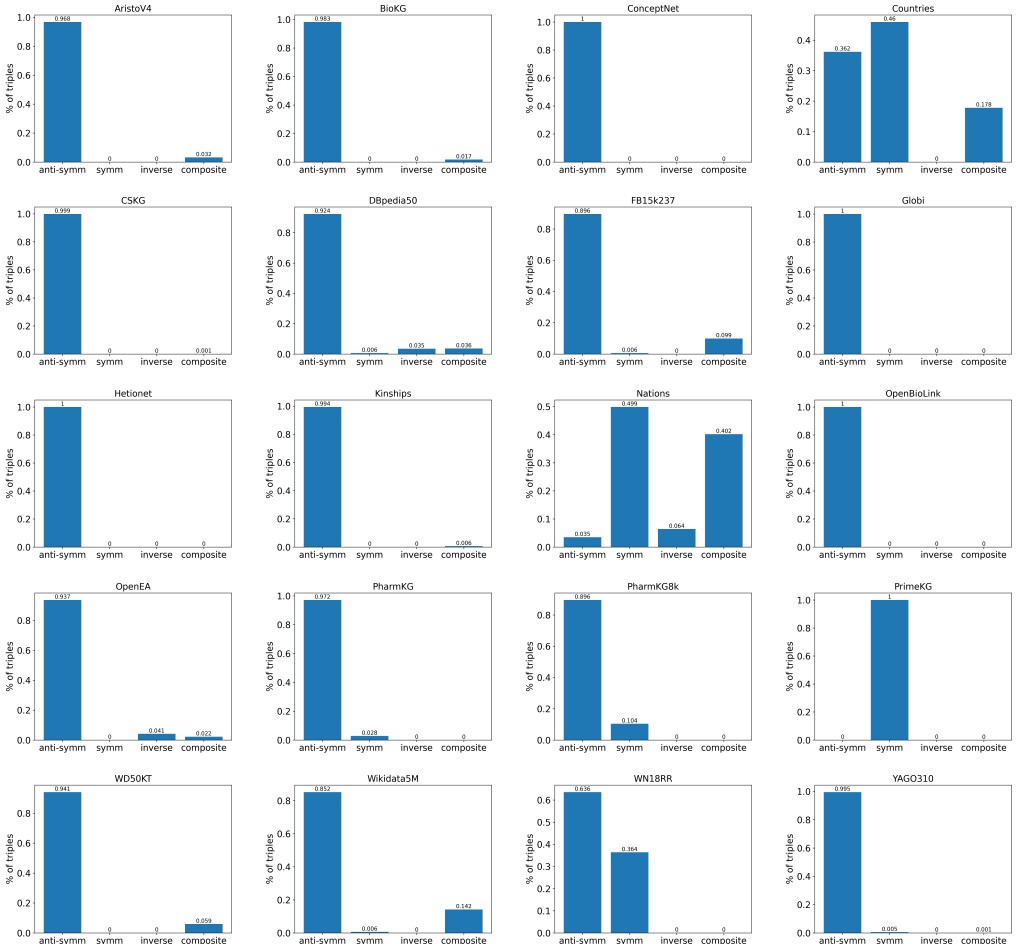

Figure 3: Proportion of (anti-)symmetric, inverse and composite relations in different KGs. Each bar is marked with the # of relations of the specified type with the $y$-axis showing the corresponding triples (as a % of all triples). Due to space limits, the CoDExMedium dataset is shown in the Appx.

relations lead to triangles in KGs and intuitively imply diminished reasoning pathways in KGs which may have an effect on downstream model performance Additionally, the distinct presence of composite relations in FB15k237 (one of the most frequently used datasets in NLP research) may lead to flaws in KG-based NLP model evaluation, unless performance on a variety of other datasets is also considered.

- Breit et al. (2020) hypothesize that the size of biomedical KGs tends to be large, which calls into double whether model results from smaller datasets are informative. We argue that beyond size, practitioners should consider the role KG properties and structural patterns during the design and testing of hypotheses and model development.
- We believe that analyzing the properties and structure of existing KGs can also benefit the future design of more robust KG datasets which incorporate diversity along different dimensions, such as the ones explored in this paper.

In conclusion, our study has implications for the broader KGs use in research: given the proliferation of models (KG link prediction, EA, LM-as-KG evaluation) across domains (natural sciences, medicine, ML and other disciples), it is worth investigating whether (and how) structural patterns, as well as their inter-domain variability across KGs, may correlate or influence KG model performance. Given the scope and scale of such an investigation, we leave it for a follow up study and encourage others – within the ML, NLP and the 'ML for sciences' communities – to further explore this topic.

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

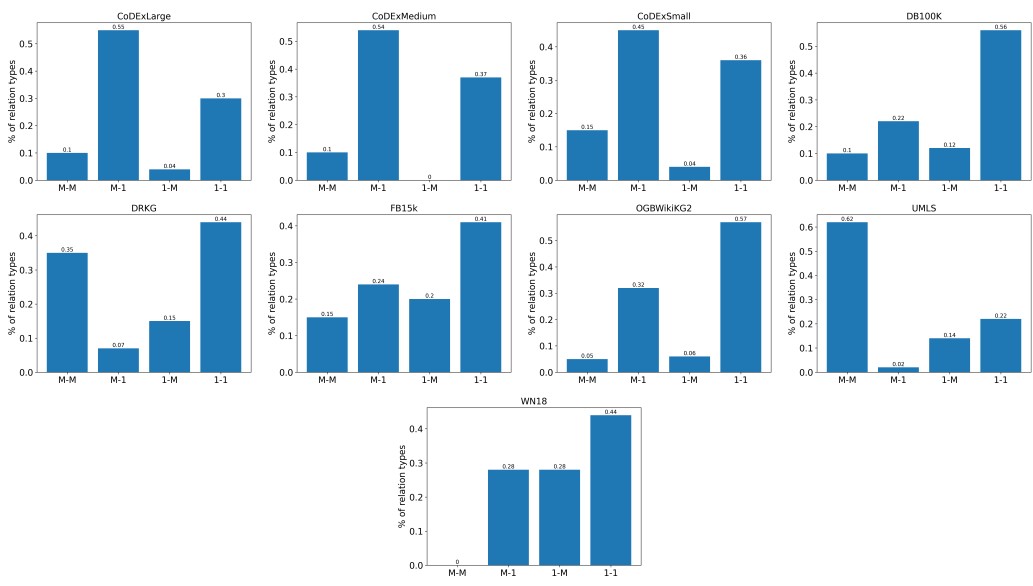

Figure 4: Supplemental panels for Fig. 2.

# A  Appendix

## A.1  Datasets used

We used all the datasets in the PyKEEN library as described in the paper with the exception of several datasets (e.g., WK3l15k, WK3l120k, CN3l, and CKG) whose underlying files are no longer available for download on the URLs the library points to.

## A.2  Degree plots

The Nations and Kinship datasets were not included in Fig. 1 due to the high number of high degree nodes in them which leads to plot scaling issues of the remaining 26 datasets. The Nation's 14 entities have degrees in the range $146 - 514$; the Kinship's entities have degrees in the $192 - 206$ range. For similar reasons we exclude the highest-degree entity (men) of the ConceptNet dataset in the plot in Fig. 1.

## A.3  Relation types, cardinalities, and metapaths

Relationship type determination, i.e. whether a relation is (anti)-symmetric, inverse, composite, is based on association rule mining. The relation classifications are based on checking whether the corresponding rules hold with sufficient support and confidence – we calculated the support using a confidence of $95\%$. We used the reference implementation in available in PyKEEN Ali et al. (2021b). Note that a relation can be of several different types.

Relation cardinality is computed similarly to the relation type.

Metapath lengths are approximated by sampling (uniformly at random) an entity $e$ from each KG and counting all the paths of length $2, 3$ and $4$ originating from $e$. Each KG was sampled 3 times, so the metapath numbers reported in Fig. 1 (right) are averaged over 3 independent entity samples, for each KG.

## A.4  Additional plots for Fig. 2 and Fig. 3

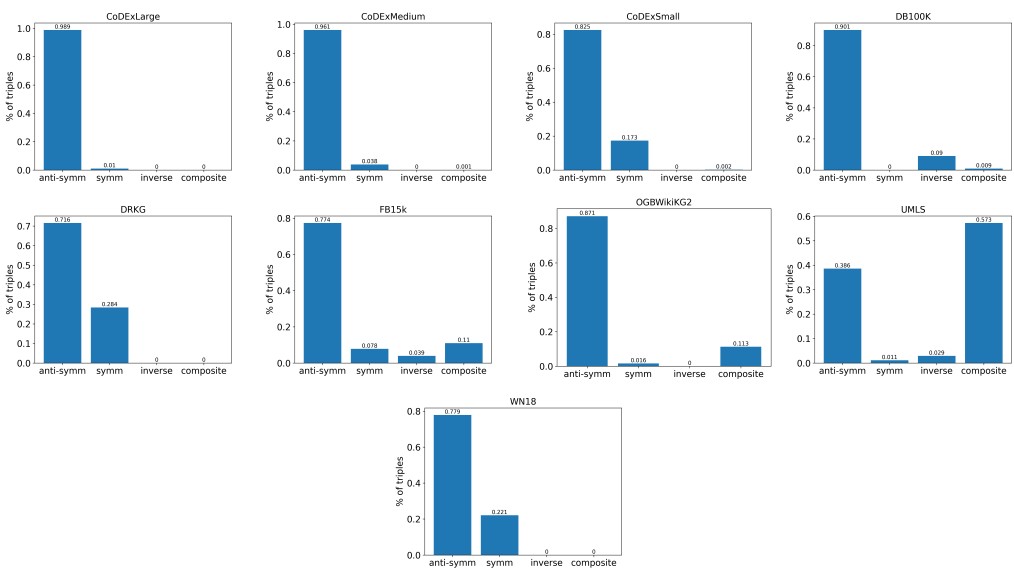

Figure 5: Supplemental panels for Fig. 3

