# OpenReview forum: "Exploring the Properties and Structure of Real Knowledge Graphs across Scientific Disciplines"
_NeurIPS.cc/2023/Workshop/AI4Science — NeurIPS 2023 Workshop AI4Science Withdrawn Submission_

### Official Review · Reviewer_W9jx · 2023-10-20
**A paper about KG study for scientific data**

**Rating:** 6
**Confidence:** 4

**Review:**

This paper proposes an evaluation for the knowledge graph in the scientifc research. They believe that their conclusions may guide users to design KG based model in a better approach. However, I have some concerns about this manuscript and evaluation pipeline:

1. The metrics used by the authors are lack of supports. It is hard for me to linke the performance of KG model on specific problems with the metrics the author used in this manuscript. For examlpe, is it possible to improve the node classification performance based on finding more number of relational patterns? I believe such question is more valuable for scienfic research.

2. It is also important to consider the certainty of KG graph edge. Some edges in current KG graph may not be found in every dataset related to this knowledge domain. Is it possible to find the certainty of each edge in one KG?

3. It seems that the author do not discuss too much about the direction in some KGs (for exmaple, cite network or regulatory network). Will the direction also affect a lot for KG based model? I think this question is also interesitng.

Therefore, I believe the authors may need more exploration for the datasets they collected, but I am happy to see their contribution.

---

### Meta-Review · Area_Chair_4rBN · 2023-10-27

**Recommendation:** Accept (Poster)
**Confidence:** 3

**Metareview:**

The paper proposes an evaluation for knowledge graphs in scientific research with the aim of guiding users in designing knowledge graph-based models more effectively. Reviewer expresses several concerns regarding the metrics used, the consideration of certainty in KG graph edges, and the impact of direction in KGs. However, the submission has its own merits, and considered sufficient as a workshop publication.